# Trematodes of Small Mammals (Erinaceomorpha, Soricomorpha, Rodentia and Chiroptera) in the Middle Volga Region (Russia)

Nadezhda Yu. Kirillova [1], Alexander A. Kirillov [1], Victoria A. Vekhnik [1], Sergei V. Shchenkov [2], Alexander I. Fayzulin [1,*] and Alexander B. Ruchin [3]

1   Institute of Ecology of Volga River Basin of RAS, Samara Federal Research Center of RAS, Togliatti 445003, Russia; nadinkirillova2011@yandex.ru (N.Y.K.); parasitolog@yandex.ru (A.A.K.); ivavika@rambler.ru (V.A.V.)
2   Department of Invertebrate Zoology, Saint Petersburg State University, St. Petersburg 199034, Russia; svshchenkov@yandex.ru
3   Joint Directorate of the Mordovia State Nature Reserve and National Park "Smolny", Saransk 430005, Russia; ruchin.alexander@gmail.com
*   Correspondence: alexandr-faizulin@yandex.ru

**Abstract:** In this study, we present our dataset containing up-to-date information about occurrences of trematodes in small mammals in the Middle Volga region (European Russia). The dataset summarizes micromammals' trematode occurrences obtained by long-term field helminthological studies of soricomorphs, erinaceomorphs, bats and rodents during a period of more than 20 years (1999–2022). Our studies of trematodes in micromammals were conducted using the method of complete helminthological necropsy. The dataset includes 7470 records of trematode occurrences in micromammals with 4483 digenean records in Samara Oblast, 2986 records in Republic of Mordovia and one trematode record in Ulyanovsk Oblast. Our dataset presents the data on 43 trematode species from 21 genera and 9 families found in the region studied. The data on trematodes from 28 species of micromammals belonging to 14 genera are presented. In total, the number of collected trematode specimens in our dataset is 153,050. Each occurrence record contains the trematode species name, basis of record, locality of finding, host species, site in host, date and authors of the record and species identification. All occurrence records are georeferenced. The dataset is based on the research of the staff of the Institute of Ecology of the Volga River basin of RAS and the Joint Directorate of the Mordovia Nature Reserve and National Park "Smolny". The distribution and diversity of trematodes of small mammals in the Middle Volga region has not been completely studied, and further investigation may reveal both new occurrences of trematodes and new host records.

**Dataset:** https://doi.org/10.15468/gmt9ct.

**Dataset License:** Creative Commons Attribution (CC-BY) 4.0 License.

**Keywords:** biodiversity; digeneans; European Russia; occurrence dataset; occurrence records; micromammals; Trematoda

## 1. Summary

The study of the diversity and distribution of helminths of vertebrates is of great importance, especially in relation to pathogenic parasites. Some trematode species can parasitize humans and vertebrates, causing serious harm to human health and significant economic damage to human economic activity. The study of helminths in mammals and digeneans in particular has always been of not only theoretical but also great practical interest for scientists, because they (mammals) serve as hosts for a number of trematodes, the causative agents of the helminthiasis of pet, farm and game animals [1–8].

Nowadays, there are almost no natural ecosystems left in the Middle Volga region that have not been affected by anthropogenic activity [3]. The deterioration of the ecological situation leads to a violation of the living conditions of invertebrates and vertebrates, a change or destruction of historically established relationships between parasites and their hosts and the involvement of humans in previously unusual host–parasite systems [3,9].

The parasitological potential of the Middle Volga region is very high, since the region has a great diversity of fauna of invertebrates and vertebrates (probable intermediate, paratenic and final hosts of trematodes) with high abundance and population density. The hydrological conditions in the area also play an important role in the implementation of the trematode life cycles, since the Middle Volga region is rich in numerous water bodies [3,10,11].

Our helminthological study of vertebrates in the Middle Volga region began in 1999. For more than 20 years of research, we have studied the trematode fauna of 28 species of small mammals (out of 85 inhabiting the region) from four orders [3,12–19]. Research was carried out in the territory of the Republic of Mordovia, Samara and Ulyanovsk Oblasts.

The main checklist of trematode species in the small mammals from the Middle Volga region was published in 2012, and it included our data on 29 species of trematodes in small mammals (insectivores, hedgehogs, bats and small rodents) [3]. The study of the parasite fauna of vertebrates in the Middle Volga region continued. And since then, new data on trematodes of small mammals have appeared. The purpose of our work is to describe the recent fauna of trematodes in small mammals in the Middle Volga region based on a novel published dataset [20].

## 2. Data Description

### 2.1. Structure of Dataset

The dataset includes 7470 records of trematode occurrences in small mammals (Erinaceomorpha, Soricomorpha, Rodentia and Chiroptera) with 4483 digenean records in Samara Oblast, 2986 records in Republic of Mordovia and one trematode record in Ulyanovsk Oblast. In total, 153,050 trematode specimens were counted in our dataset.

In our dataset, each trematode occurrence includes basic information on the location (latitude/longitude), date of occurrence, names of the observer and identifier. The geographical coordinates were determined in the study site with the help of a GPS device or after field works using Google Maps (Table 1).

**Table 1.** Description of the data in the dataset.

| Column Label | Column Description |
| --- | --- |
| occurrenceID | An identifier for the occurrence (as opposed to a particular digital record of the occurrence) |
| basisOfRecord | The specific nature of the data record: preservedSpecimen |
| scientificName | The full scientific name, including the genus name and the lowest level of taxonomic rank with the authority |
| nameAccordingTo | The reference to the source in which the specific taxon concept circumscription is defined or implied |
| kingdom | The full scientific name of the kingdom in which the taxon is classified |
| phylum | The full scientific name of the phylum or division in which the taxon is classified |
| class | The full scientific name of the class in which the taxon is classified |
| family | The full scientific name of the family in which the taxon is classified |
| lifeStage | The age class or life stage of the Organism(s) at the time the Occurrence was recorded |
| higherTaxon | A list (concatenated and separated) of the names for the taxonomic ranks less specific than the ScientificName: superfamily |
| decimalLatitude | The geographic latitude of location in decimal degree |
| decimalLongitude | The geographic longitude of location in decimal degrees |
| geodeticDatum | The ellipsoid, geodetic datum or spatial reference system (SRS) upon which the geographic coordinates given in decimalLatitude and decimalLongitude are based |
| location | A spatial region or named place |

**Table 1.** *Cont.*

| Column Label | Column Description |
| --- | --- |
| stateProvince | The name of the next smaller administrative region than country (state, province, canton, department, region, etc.) in which the Location occurs |
| country | The name of the country in which the Location occurs |
| countryCode | The standard code for the country in which the Location occurs |
| individualCount | The number of individuals represented present at the time of the Occurrence |
| higherGeography | A list (concatenated and separated) of geographic names less specific than the information captured in the locality term |
| eventDate | The date when material from the trap was collected or the range of dates during which the trap collected material |
| year | The integer day of the month on which the Event occurred |
| month | The ordinal month in which the Event occurred |
| day | The integer day of the month on which the Event occurred |
| habitat | A category or description of the habitat in which the Event occurred: site in host |
| associatedTaxa | A list (concatenated and separated) of identifiers or names of taxa and the associations of this Occurrence to each of them: host |
| recordedBy | A person, group, or organization responsible for recording the original Occurrence |
| identifiedBy | A list of names of the people who assigned the taxon to the subject |

*2.2. Dataset Description*

The dataset contains data on 43 species of Trematoda from 21 genera and 9 families found in the Republic of Mordovia, Ulyanovsk Oblast and Samara Oblast and documented simultaneously with coordinates (Figure 1).

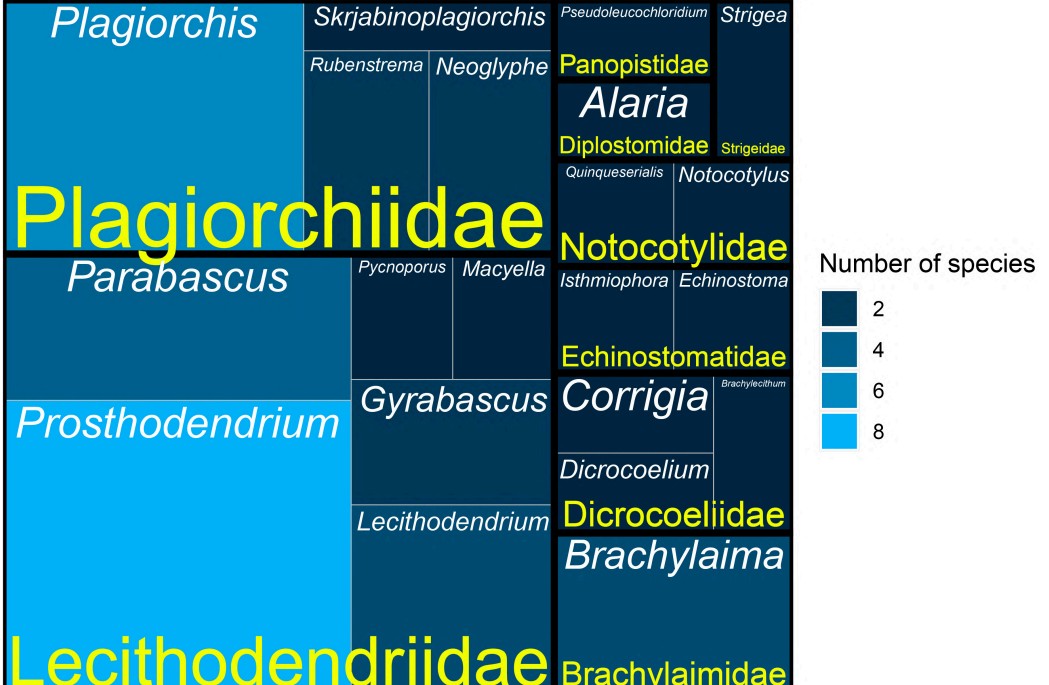

**Figure 1.** Taxonomic distribution of trematodes amongst families in the dataset.

The families Lecithodendriidae (19) and Plagiorchiidae (11) are the largest in terms of species diversity (Figure 1). This is 69.8% of whole number of trematode species found in small mammals from the Middle Volga region. The families Brachylaimidae (3 species), Dicrocoeliidae (3), Echinostomatidae (2) and Notocotylidae (2) are much less represented. The families Panopistidae, Strigeidae and Diplostomatidae are found in the small mammals of the Middle Volga region by only one species each.

In terms of number of species, the five genera of trematodes *Prosthodendrium* (8 species), *Plagiorchis* (6), *Parabascus* (4), *Lecithodendrium* (3) and *Brachylaima* (3) prevail, representing 55.8% of all digenean species (Figure 1). Other genera of trematodes are represented by one or two species.

The total number of trematode specimens in our dataset is 153,050. The number of trematode occurrences of different families and genera is shown in Figure 2.

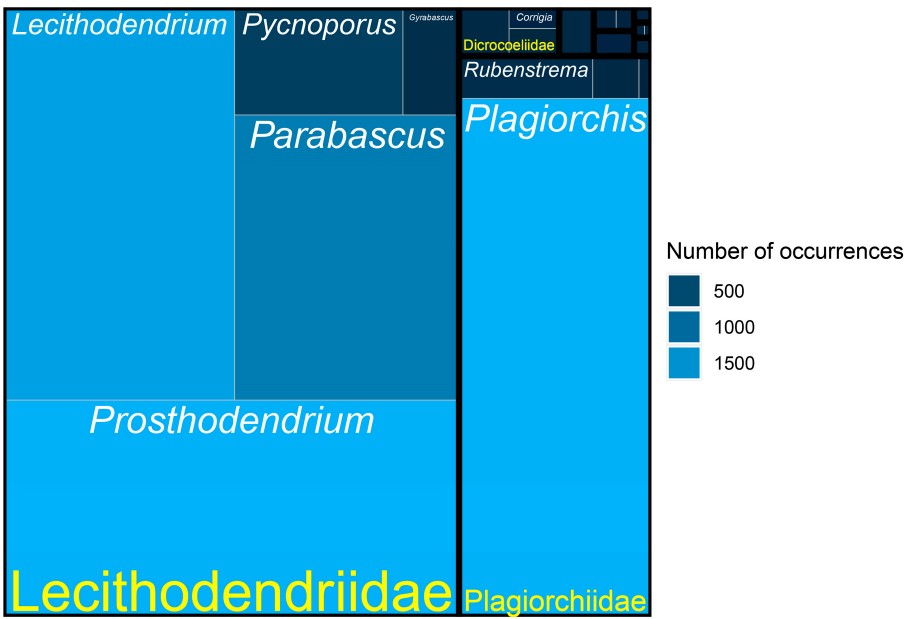

**Figure 2.** Distribution of trematode occurrences amongst families in the dataset.

Here, the same families Lecithodendriidae (5233) and Plagiorchiidae (2060) are also in the lead, accounting for 97.6% of all occurrences (Figure 2). Moreover, 73.4% is accounted for by three genera from these families: *Plagiorchis* (25.4%, 1897 occurrences), *Prosthodendrium* (25.0%, 1874) and *Lecithodendrium* (23.0%, 1714).

We found 38 species of trematodes in micromammals in the territory of the Republic of Mordovia and 31 digenean species in animals from the Samara Oblast. Twenty-six species of trematodes are common in the Samara Oblast and the Republic of Mordovia. Twelve species of trematodes are found only in the mammals of Mordovia; five species of digeneans are found only in Samara Oblast. From the territory of the Ulyanovsk Oblast, only *Pipistrellus kuhlii* was studied, in which one species of trematodes, *Parabascus semisquamosus*, was noted. This trematode is common to all three studied areas of the Middle Volga region (Figure 3).

In our dataset, the data on trematodes in 28 species of micromammals belonging to 14 genera are presented. Members of the order Chiroptera have the greatest number of trematode species—23 (Figure 4).

The trematode fauna of mammals of the orders Rodentia (13) and Soricomorpha (8) is less diverse. Only two species of trematodes were found in the only studied member of the order Erinaceomorpha, *Erinaceus roumanicus*. In mammals from four studied orders, no common species was revealed (Figure 4). In bats and rodents, one common species, *P. elegans*, has been noted; in hedgehogs and soricomorphs—*Strigea strigis*, mtc., as well as in rodents and hedgehogs—*Istmiophora melis* (Figure 4).

Among bats, according to the number of trematode species in our database, the leading species are *Nyctalus noctula* (12 species), *Myotis daubentonii*, *Myotis brandtii* and *Eptesicus nilssoni* (11 species each); *Nyctalus leisleri* and *Vespertilio murinus* (10 species each); *Myotis dasycneme* and *Pipistrellus nathusii* (9 species each). The trematode fauna is less diverse in *Myotis nattereri* and *M. mystacinus* with six species each, *Plecotus auritus* and *Pipistrellus kuhlii* with five species each, and *Pipistrellus pygmaeus* with four species. The smallest

number of trematode species was noted in *Nyctalus lasiopterus* (2) and *Eptesicus serotinus* (1), which is mainly caused by investigation of single specimens of these bat species.

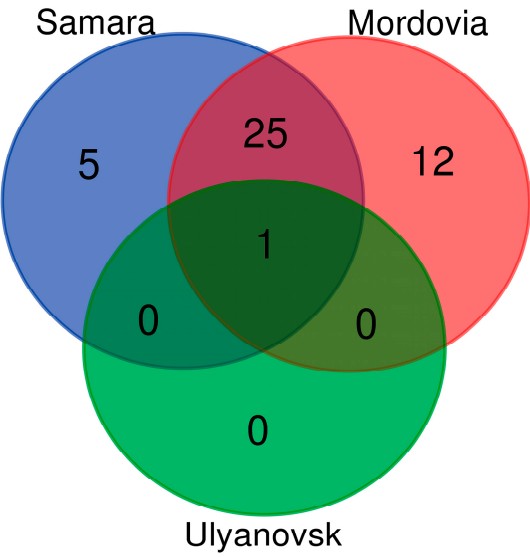

**Figure 3.** Number of trematode species in small mammals from different areas of the Middle Volga region. Blue circle—Samara Oblast, red circle—Republic of Mordovia, green circle—Ulyanovsk Oblast. In over-lapping areas, the number of common trematode species is shown.

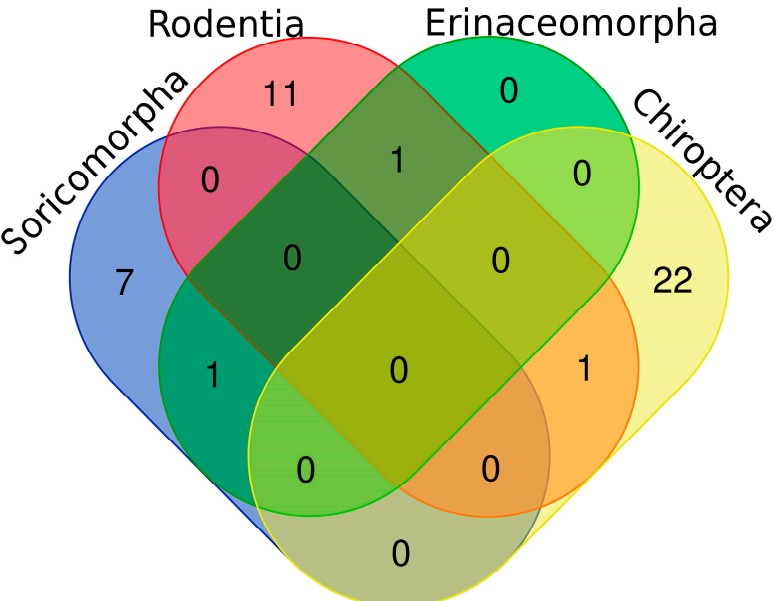

**Figure 4.** Number of trematode species in members of different orders of small mammals. Blue circle—shrews, Soricomorpha, red circle—rodents, Rodentia, green circle—hedgehogs, Erinaceo-morpha, yellow circle—bats, Chiroptera. The number of common trematode species are shown in overlapping areas.

Among rodents, *Clethrionomys glareolus* leads in terms of the number of species, in which six species of trematodes were found. Trematodes are less represented in *Apodemus agrarius Apodemus uralensis* (five species each), and *Apodemus flavicollis* (four). Two species of trematodes were found in *Microtus cf arvalis*, and one each in *Microtus oeconomus*, *Mus musculus* and *Rattus norvegicus*.

Among soricomorphs, the largest number of trematode species was found in *Sorex araneus*—six species. *Neomys fodiens* has four species, and *Sorex minutus* has three. Only one species of digeneans was found in *Neomys anomalus*, which is again due to the study of a single specimen of this species.

The data of our database show that *Plagiorchis koreanus* has the largest number of host species in the Middle Volga region—12 species (Table 2).

**Table 2.** Host species of trematodes in the dataset. In decreasing order of host species number.

| Parasite | Host Species |
|---|---|
| *Plagiorchis koreanus* (Ogata, 1938) | *Myotis brandtii, M. daubentonii, M. dasycneme, M. nattereri, M. mystacinus, Plecotus auritus, Vespertilio murinus, Eptesicus nilssoni, Nyctalus noctula, N. leisleri, Pipistrellus nathusii, P. kuhlii* |
| *Plagiorchis elegans* (Rudolphi, 1802) | *M. daubentonii, N. noctula, V. murinus, P. auritus, Apodemus agrarius, A. flavicollis, A. uralensis, Clethrionomys glareolus* |
| *Lecithodendrium linstowi* Dollfus, 1931 | *M. brandtii, M. daubentonii, M. mystacinus, Eptesicus nilssoni, N. noctula, N. leisleri, P. nathusii, Pipistrellus kuhlii* |
| *Prosthodendrium chilostomum* (Mehlis, 1831) | *M. brandtii, M. daubentonii, M. dasycneme, M. nattereri, M. mystacinus, P. auritus, E. nilssoni, N. leisleri* |
| *Parabascus semisquamosus* (Braun, 1900) | *Eptesicus serotinus, E. nilssoni, Nyctalus lasiopterus, N. noctula, N. leisleri, Pipistrellus pygmaeus, P. nathusii, P. kuhlii* |
| *Pycnoporus heteroporus* (Dujardin, 1845) | *V. murinus, E. nilssoni, N. noctula, P. nathusii, P. kuhlii, P. pygmaeus* |
| *Prosthodendrium ascidia* (van Beneden, 1873) | *M. brandtii, M. daubentonii, M. dasycneme, M. nattereri, M. mystacinus, P. nathusii* |
| *Prosthodendrium ilei* Zdzitowiecki, 1969 | *V. murinus, E. nilssoni, N. noctula, N. leisleri, P. nathusii, P. kuhlii* |
| *Lecithodendrium rysavyi* Dubois, 1960 | *V. murinus, E. nilssoni, N. noctula, N. leisleri, P. nathusii, P. pygmaeus* |
| *Dicrocoelium dendriticum* (Rudolphi, 1819) | *C. glareolus, A. agrarius, A. flavicollis, A. uralensis, Mus musculus* |
| *Plagiorchis muelleri* Tkach & Sharpilo, 1990 | *M. brandtii, M. mystacinus, M. nattereri, V. murinus, N. noctula* |
| *Parabascus duboisi* (Hurkova, 1961) | *M. brandtii, Myotis daubentonii, Myotis dasycneme, M. nattereri, M. mystacinus* |
| *Parabascus lepidotus* Looss, 1907 | *V. murinus, E. nilssoni, N. noctula, N. leisleri, N. lasiopterus* |
| *Lecithodendrium skrjabini* Matsaberidze, 1963 | *V. murinus, N. noctula, N. leisleri, P. nathusii, P. pygmaeus* |
| *Plagiorchis mordovii* Schaldybin, 1958 | *M. brandtii, M. daubentonii, M. dasycneme, V. murinus* |
| *Rubenstrema exasperatum* (Rudolphi, 1819) | *Sorex araneus, Sorex minutus, Neomys fodiens, Neomys anomalus* |
| *Plagiorchis vespertilionis* (Muller, 1784) | *M. brandtii, M. daubentonii, M. dasycneme* |
| *Prosthodendrium hurkovaae* Dubois, 1960 | *M. daubentonii, M. dasycneme, P. auritus* |
| *Prosthodendrium longiforme* (Bhalerao, 1926) | *M. brandtii, M. daubentonii, P. auritus* |
| *Gyrabascus amphoraeformis* (Modlinger, 1930) | *M. brandtii, M. daubentonii, M. dasycneme* |
| *Gyrabascus oppositus* (Zdzitowiecki, 1969) | *N. noctula, N. leisleri, P. nathusii* |
| *Brachylaima recurva* (Dujardin, 1845) | *C. glareolus, Microtus oeconomus* |
| *Corrigia vitta* (Dujardin, 1845) | *A. flavicollis, A. uralensis* |
| *Isthmiophora melis* (Schrank, 1788) | *A. agrarius, Rattus norvegicus* |
| *Neoglyphe sobolevi* (Shaldybin, 1953) | *S. araneus, S. minutus* |
| *Notocotylus noyeri* Joyeux, 1922 | *C. glareolus, Microtus cf arvalis* |
| *Parabascus magnitestis* Khotenovski, 1985 | *E. nilssoni, N. noctula* |
| *Prosthodendrium cryptolecithum* Zdzitowiecki, 1969 | *M. brandtii, M. dasycneme* |
| *Prosthodendrium megacotyle* Ogata, 1939 | *E. nilssoni, N. leisleri* |
| *Prosthodendrium skrjabini* (Shaldybin, 1948) | *V. murinus, E. nilssoni* |

**Table 2.** *Cont.*

| Parasite | Host Species |
| --- | --- |
| *Pseudoleucochloridium soricis* (Soltys, 1952) | *S. araneus, N. fodiens* |
| *Skrjabinoplagiorchis polonicus* (Soltys, 1957) | *A. flavicollis, A. uralensis* |
| *Strigea strigis* (Schrank, 1788), mtc. | *S. araneus, Erinaceus roumanicus* |
| *Alaria alata* (Goeze, 1782), msc. | *S. araneus* |
| *Brachylaima aequans* (Looss, 1899) | *A. agrarius* |
| *Brachylaima fulvum* Dujardin, 1843 | *S. araneus* |
| *Brachylecithum rodentini* Agapova, 1955 | *C. glareolus* |
| *Echinostoma miyagawai* Ischii, 1932 | *A. agrarius* |
| *Macyella apodemi* Jourdane & Triquell, 1973 | *A. uralensis* |
| *Neoglyphe locellus* (Kossack, 1910) | *N. fodiens* |
| *Plagiorchis arvicolae* Schulz & Skvorzov, 1931 | *C. glareolus* |
| *Rubenstrema opisthovitellinus* (Soltys, 1954) | *N. fodiens* |
| *Quinqueserialis wolgaensis* Skvortsov, 1935 | *M. cf arvalis* |

Four trematode species (*Plagiorchis elegans*, *Lecithodendrium linstowi*, *Prosthodendrium chilostomum* and *Parabascus semisquamosus*) each have eight species of hosts. The number of hosts varies from 2 to 6 species in 28 species of trematodes. Ten species of digeneans have only one host species each (Table 2).

Half of the trematode species found in micromammals of the Middle Volga region has a Palearctic distribution—21 species. The Holarctic faunistic complex includes six species of digeneans. Five species of parasites have a cosmopolitan distribution. The distribution range of 11 trematodes is limited to Europe.

We have obtained new data on the fauna of trematodes in micromammals. Two species of trematodes were noted in small mammals for the first time in Russia: *Prosthodendrium cryptolecithum* Zdzitowiecki, 1969, and *Macyella apodemi* Jourdane & Triquell, 1973. For the first time in the Middle Volga region, five species of trematodes were revealed: *Brachylaima aequans* (Looss, 1899), *Brachylaima recurva* (Dujardin, 1845)*, Rubenstrema opisthovitellinus* (Soltys, 1954), *Gyrabascus amphoraeformis* (Modlinger, 1930) and *Gyrabascus oppositus* (Zdzitowiecki, 1969). Thus, the list of trematodes of small mammals in the Middle Volga region, taking into account the literature data, includes 51 species of digeneans. [3,12–19,21,22], this study.

In addition, we have added the list of trematode hosts in the region studied. Thus, for *B. recurva*, the new host is *Microtus oeconomus*. Previously, *Echinostoma miyagawai* Ischii, 1932, was recorded in the Middle Volga region only in birds. In the Middle Volga region, we first recorded the parasite in *Apodemus agrarius*. Metacercariae of *Strigea strigis* (Schrank, 1788) were found in *Sorex araneus* and *Erinaceus roumanicus* for the first time in the Middle Volga region, although earlier they were noted only in amphibians and reptiles here. The diversity of trematodes in small mammals of the region has not been fully studied, and further research may reveal both new occurrences of trematodes and new host records of digeneans.

## 3. Methods

### 3.1. Study Area Description

The Middle Volga region is located in the east of the Russian Plain and occupies a territory approximately within 52–57° northern latitude and 45–56° eastern longitude [23] (Figure 5).

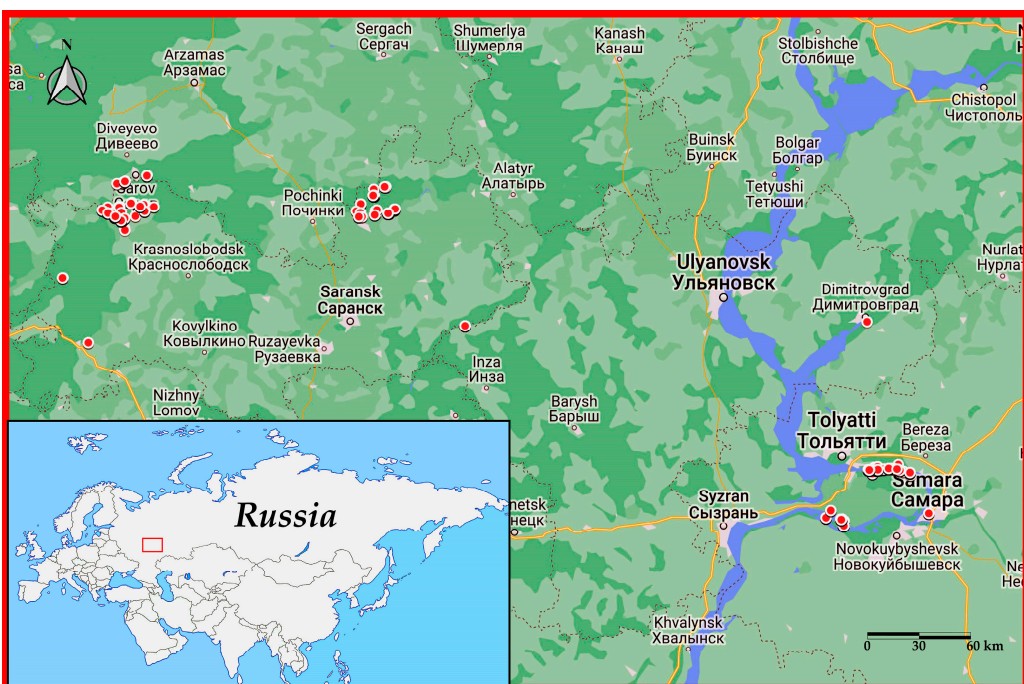

**Figure 5.** The Middle Volga region and sampling localities of trematodes from small mammals within the area studied. The locations of trematode occurrences are shown as red circles. The location of the Middle Volga region (at the minor map on left) is shown as a red frame.

Administratively, the region includes the republics of Mari El, Tatarstan and Chuvashia, Ulyanovsk and Samara Oblasts, the eastern districts of the Republic of Mordovia, the Nizhny Novgorod and Penza Oblasts, the western districts of the Republic of Bashkortostan and the Orenburg Oblast and the northern districts of the Saratov Oblast.

The relief of the Middle Volga region is relatively flat [23,24]. Wide watersheds alternate in the region with shallow river valleys, and highlands alternate with lowlands. The average altitude of the region is 150–180 m above sea level [23,24]. The climate of the region is temperate continental [23]. The hydrographic network of the Middle Volga region is well developed in the northern forest-steppe areas and sharply rarefied to the south in the steppe zone. A characteristic feature of the region is the presence of the large rivers—Kama, Belaya, Vyatka, Vetluga, Sura and Samara, which are part of the Volga river system, the largest river in Europe (3531 km) [10,11,24].

The soil coverage of the Middle Volga region is dominated by podzolic forest-steppe soils, the proportion of which is about 35% [23–25]. The Volga basin in its middle course is diverse in landscapes: coniferous and mixed forests in the north of the region and steppe in the south, and the central and largest part of the region is occupied by forest-steppe [24]. The location of the region within the forest, forest-steppe and steppe zones determines the exceptional diversity of fauna and flora. The Middle Volga region is characterized by a variety of vegetation types: deciduous, mixed and dark coniferous forests, steppes, fields and floodplain vegetation. The vegetation coverage of the Middle Volga region is represented by approximately 2050 plant species [26,27]. Furthermore, 85 species of mammals inhabit the Middle Volga region, including 10 species of soricomorphs, 3 hedgehogs, 15 bats and 38 rodents [28–36].

### 3.2. Description of Parasitological Data

The data presented in our dataset is the result of many years of research by the authors of the trematode fauna of small mammals in the Middle Volga region conducted from 1999 to 2022 [3,12–19]. The dataset is based on own results of parasitological study of vertebrates and records from own field diaries. Most of the data on the trematode fauna

in micromammals from the region studied was published earlier but without reference to geographic coordinates [3,12–19]. The geographical coordinate reference to each trematode occurrence record is given for the first time.

All geographical references were done by fixing the geographical coordinates of the sampling sites of the trematode hosts, small mammals, using a GPS Navigator or Google maps (https://www.google.ru/maps/, accessed on 14 April 2023) [37]. The accuracy of the measurement of coordinates is 10 m. The accuracy of determining coordinates is up to the fourth digit. In all records, the WGS-84 coordinate system is used.

The voucher specimens of trematodes studied are stored in the parasitological collection of the Institute of Ecology of Volga Basin of RAS (IEVB RAS), a branch of the Samara Federal Research Center of the Russian Academy of Sciences. The essential part of trematode samples is stored in 96% ethanol in the personal collections of A.A. Kirillov and N.Yu. Kirillova.

Data were visualized using "R" programming language [38] with "treemapify" [39] and "ggplot2" [40] packages. Venn diagrams were prepared with on-line tool, available through the following link: "https://bioinformatics.psb.ugent.be/webtools/Venn/, accessed on 15 May 2023" [41].

The trematode species were identified according to Kirillov et al. [3], Genov [42], Ryzhikov et al. [43], Khotenovsky [44–46], Odening [47], Sharpilo, Iskova [48], Zdzietowiecki [49,50], Kirillova et al. [51] and Sokolov et al. [52]. The helminth taxonomy is given according to Fauna Europaea (https://fauna-eu.org/, accessed on 3 May 2023) [53] and articles of Sokolov et al. [52], Kirillova et al. [51].

**Author Contributions:** Conceptualization, A.A.K., A.B.R. and N.Y.K.; methodology, A.A.K., N.Y.K. and V.A.V.; software, A.A.K, S.V.S. and A.I.F.; validation, A.A.K., N.Y.K., S.V.S. and V.A.V.; formal analysis, A.A.K., N.Y.K. and S.V.S.; investigation, A.A.K., N.Y.K. and V.A.V.; resources, A.A.K., A.B.R., N.Y.K. and A.I.F.; data curation, A.A.K., N.Y.K. and V.A.V.; writing—original draft preparation, A.A.K., N.Y.K. and S.V.S.; writing—review and editing, A.A.K., N.Y.K., V.A.V. and S.V.S.; visualization, A.A.K., N.Y.K. and A.B.R.; supervision, A.A.K., N.Y.K. and A.B.R.; project administration, N.Y.K., A.I.F. and V.A.V.; funding acquisition, A.A.K. and A.B.R. All authors have read and agreed to the published version of the manuscript.

**Funding:** The research in the Samara region was funded by Russian Science Foundation, grant number 23-24-10021 (A.A. Kirillov, N.Yu. Kirillova and V.A. Vekhnik). This research was partially performed within the framework of the state assignment 1-22-31-1 from Ministry of Natural Resources and Ecology of the Russian Federation and the research theme № 1021060107212-5-1.6.20; 1.6.19 "Change, sustainability and biodiversity conservation under the global climate change impact and intense anthropogenic pressure on the ecosystems of the Volga River basin" of the Institute of Ecology of the Volga River Basin, a branch of the Samara Federal Research Center of the Russian Academy of Sciences.

**Institutional Review Board Statement:** All applicable international, national and institutional guidelines for the use and care of wild animals were followed. Our research was conducted in compliance with the ethical standards of humane treatment of animals in accordance with the recommended standards described by the Directive of the European Parliament and of the Council of the European Union of 22 September 2010, "On the protection of animals used for scientific purposes" (EU Directive 2010/63/EU). The material for parasitological study was obtained as a result of long-term fieldworks on accounting for the number of wild animals. Trapping and research of small mammals was carried out in accordance with agreements on scientific cooperation with the Samarskaya Luka National Park, Zhiguli Nature Reserve in 1999–2022 and the Federal State Budgetary Institution "Reserved Mordovia" ("Zapovednaya Mordovia") in 2018–2022.

**Data Availability Statement:** Not applicable.

**Acknowledgments:** The authors are deeply grateful to Natalya Ivanova (Institute of Mathematical Problems of Biology RAS) for valuable suggestions and technical support in the production of our dataset.

**Conflicts of Interest:** The authors declare no conflict of interest.

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
