# Peer review of "Trematodes of Small Mammals (Erinaceomorpha, Soricomorpha, Rodentia and Chiroptera) in the Middle Volga Region (Russia)"

_diversity, doi:10.3390/d15070796_

Round 1

Reviewer 1 Report

The authors present a Data Descriptor paper about Trematodes of Small Mammals in the Middle Volga Region (Russia). This work is a very robust and reliable data description since it has analyzed a total of  7470 mammals and recovered a total of 153,050 trematodes from 43 different species. The sampling has happened along 20 years, proving it is a representative sample of the helminthic prevalence in small mammals in the region.

The work is very clearly presented and the English language is excellent. I have only a few comments:

- Table 1. Some of the words "event" have a capital E. Please clarify what associatedTaxa mean. Why does it say : host?

- What is the difference between Figures 1 and 2? Could you please clarify?

- Unify the format of the species name + authority, year in Table 2. Some of them have parenthesis while others don´t.

- Improve Figures quality, specially Figure 5, is very pixeled.

- It would be of great interest to analyze the interactions among the trematode species found like coinfections. I am sure more than one species was found in some mammals. 

- It would also be very interesting to see a timeline graph showing the prevalence of species found over the years to see if the trematode populations change over time or even the type of host with the highest prevalences over time, since their populations might have also varied over time.

Author Response

The authors present a Data Descriptor paper about Trematodes of Small Mammals in the Middle Volga Region (Russia). This work is a very robust and reliable data description since it has analyzed a total of 7470 mammals and recovered a total of 153,050 trematodes from 43 different species. The sampling has happened along 20 years, proving it is a representative sample of the helminthic prevalence in small mammals in the region.

Dear Reviewer,

Thank you for your interest and valuable comments on our work.

The work is very clearly presented and the English language is excellent. I have only a few comments:

- Table 1. Some of the words "event" have a capital E. Please clarify what associatedTaxa mean. Why does it say: host?

The spelling of “Event” in the Table 1 has been corrected.

All column names in the GBIF database are strictly regulated and are strictly taken from the List of Darwin Core terms (Term indices) (https://dwc.tdwg.org/list/).

associatedTaxa – Term index proposed by GBIF curator Natalia Ivanova.

This is the only Term index that would be suitable for characterizing the hosts of trematodes.

“host” in the associatedTaxa description is a clarification of what type of associated organism for this trematode species is given.

- What is the difference between Figures 1 and 2? Could you please clarify?

Figure 1 shows the species diversity (number of trematode species) of trematode families (and distinct genera) in our database. Figure 2 shows the total number of occurrence records of the same trematode taxa in the database.

- Unify the format of the species name + authority, year in Table 2. Some of them have parenthesis while others don´t.

All captions in the Table 2 are made according to the rules of the Code of International Nomenclature (see Fauna Europaea https://fauna-eu.org/). According to the rules, it is enough to cite only the author (authors) who first described the species. If after that the taxonomic status of the species changed (moved to another genus by other authors), then the first described author and year are put in brackets. Names of parasites, where authors without parentheses means that the status of this species did not change after the original description.

- Improve Figures quality, specially Figure 5, is very pixeled.

In our opinion, the quality of the drawings is good. Figures 1-4 were made in the R software and transferred to the article. We cannot change them.

Figure 5 (map) in the article serves only to schematically show where the research sites were located in the Middle Volga region. The detailed high resolution map is available in our database: https://www.gbif.org/dataset/25dee5e7-4e48-49a3-987f-a6799c9ed568

- It would be of great interest to analyze the interactions among the trematode species found like coinfections. I am sure more than one species was found in some mammals.

We came up with this idea after consultations with the curator of our database, Natalya Ivanova, but after the publication of this database on trematodes in GBIF. For such an analysis, it was first necessary to register a database on the occurrences of small mammals hosting parasites in the study area. And then link the data on trematodes with each specific host individual with reference to geographic coordinates.

Therefore, this work is only a description of the database of trematode sightings registered in the GBIF. In future work, we will try to take into account your suggestion and do something similar.

- It would also be very interesting to see a timeline graph showing the prevalence of species found over the years to see if the trematode populations change over time or even the type of host with the highest prevalences over time, since their populations might have also varied over time.

This article is a description of a database that does not imply an ecological analysis of trematodes in a similar article. But similar timeline graphs can be found for every trematode species in our database. For example, forb Brachylecithum rodentini Agapova, 1955: https://www.gbif.org/occurrence/charts?q=Brachylecithum%20rodentini%20Agapova,%201955&dataset_key=25dee5e7-4e48-49a3-987f-a6799c9ed568&advanced=1

Reviewer 2 Report

This scientific work is a great contribution to the data on the biodiversity of trematodes in certain small mammals. A large number of samples were processed, which gives the significance of the results.

Author Response

This scientific work is a great contribution to the data on the biodiversity of trematodes in certain small mammals. A large number of samples were processed, which gives the significance of the results.

Dear Reviewer,

Thank you for high appreciating our work.

This scientific work is a great contribution to the data on the biodiversity of trematodes in certain small mammals. A large number of samples were processed, which gives the significance of the results.

Dear Reviewer,

Thank you for high appreciating our work.

Reviewer 3 Report

Diversity: Trematodes of Small Mammals (Erinaceomorpha, Soricomorpha, Rodentia and Chiroptera) in the Middle Volga Region (Russia).

In addition to providing valuable information about Russian digenean fauna, the manuscript includes updated information, making it even more relevant. The graphics are well presented, the openly accessible dataset is user-friendly, and contains the information mentioned in the present study. The authors also add updated information on the geographical distribution of previous publications. It is without a doubt a very important research resource not only for locals, but also for researchers worldwide. I have outlined a few points for the authors to consider below.

1. It is not specified which material is deposited in a permanent repository - at the Institute of Ecology of Volga Basin of RAS (IEVB RAS) - and which material is deposited in a personal collection. Can anyone revaluate the material deposited in their personal collection? Is everything deposited or not? In parasites, this is especial important. Here are listed a few papers highlighting the dire need for people to deposit all the materials they collect.

Thompson CW, Phelps KL, Allard MW, Cook JA, Dunnum JL, Ferguson AW, Gelang M, Khan FAA, Paul DL, Reeder DM, Simmons NB, Vanhove MPM, Webala PW, Weksler M, Kilpatrick CW. 2021. Preserve a voucher specimen! The critical need for integrating natural history collections in infectious disease studies. mBio 12:e02698-20. https://doi.org/10.1128/mBio.02698-20.

Non-repeatable science: assessing the frequency of voucher specimen deposition reveals that most arthropod research cannot be verified. Shaun Turney, Elyssa R. Cameron, Christopher A. Cloutier,and Christopher M. Buddle

Building an integrated infrastructure for exploring biodiversity: field collections and archives of mammals and parasites.  Kurt E Galbreath,1 Eric P Hoberg,2 Joseph A Cook,2 Blas Armién,3 Kayce C Bell,4 Mariel L Campbell,2 Jonathan L Dunnum,2 Altangerel T Dursahinhan,6 Ralph P Eckerlin,5 Scott L Gardner,6 Stephen E Greiman,7 Heikki Henttonen,8 F Agustín Jiménez,9 Anson V A Koehler,10 Batsaikhan Nyamsuren,11 Vasyl V Tkach,12 Fernando Torres-Pérez,13 Albina Tsvetkova,14 and Andrew G Hope15

Hoberg E.P. Foundations for an integrative parasitology: collections, archives, and biodiversity informatics. Comp. Parasitol. 2002; 69: 124-131

2. Taking a count that the present manuscript contains new records, did the material collection follow any ethical procedures? This information could be included by the authors.

3. I missed the literature regarding parasite identification in the updated records. All parasites were identified at the species level. Despite being a database, I think that for the updated records, there could be a paragraph talking about species identification; the key used, and which characters were observed which agrees with described species.

Author Response

In addition to providing valuable information about Russian digenean fauna, the manuscript includes updated information, making it even more relevant. The graphics are well presented, the openly accessible dataset is user-friendly, and contains the information mentioned in the present study. The authors also add updated information on the geographical distribution of previous publications. It is without a doubt a very important research resource not only for locals, but also for researchers worldwide.

Dear Reviewer,

Thank you for your interest and valuable comments on our work.

I have outlined a few points for the authors to consider below.

  1. It is not specified which material is deposited in a permanent repository - at the Institute of Ecology of Volga Basin of RAS (IEVB RAS) - and which material is deposited in a personal collection. Can anyone revaluate the material deposited in their personal collection? Is everything deposited or not? In parasites, this is especial important. Here are listed a few papers highlighting the dire need for people to deposit all the materials they collect.

Thompson CW, Phelps KL, Allard MW, Cook JA, Dunnum JL, Ferguson AW, Gelang M, Khan FAA, Paul DL, Reeder DM, Simmons NB, Vanhove MPM, Webala PW, Weksler M, Kilpatrick CW. 2021. Preserve a voucher specimen! The critical need for integrating natural history collections in infectious disease studies. mBio 12:e02698-20. https://doi.org/10.1128/mBio.02698-20.

Non-repeatable science: assessing the frequency of voucher specimen deposition reveals that most arthropod research cannot be verified. Shaun Turney, Elyssa R. Cameron, Christopher A. Cloutier, and Christopher M. Buddle Building an integrated infrastructure for exploring biodiversity: field collections and archives of mammals and parasites. 

Kurt E Galbreath, Eric P Hoberg, Joseph A Cook, Blas Armién, Kayce C Bell, Mariel L Campbell, Jonathan L Dunnum, Altangerel T Dursahinhan, Ralph P Eckerlin, Scott L Gardner, Stephen E Greiman, Heikki Henttonen, F Agustín Jiménez, Anson V A Koehler, Batsaikhan Nyamsuren, Vasyl V Tkach, Fernando Torres-Pérez, Albina Tsvetkova, and Andrew G Hope

Hoberg E.P. Foundations for an integrative parasitology: collections, archives, and biodiversity informatics. Comp. Parasitol. 2002; 69: 124-131

Necessary changes have been made to the manuscript.

The voucher specimens of trematodes are stored in the parasitological collection of the Institute of Ecology of Volga Basin of RAS (IEVB RAS). And they are available to fellow parasitologists.

In our personal collection, trematodes are mainly stored in 70 and 96% ethanol and are used by the authors for further morphological or molecular studies. As well as total mounts of rare and solitary trematodes. For example, Macyella apodemi, in our fieldworks met only once in the amount of 3 specimens: 2 specimens – we made total mount (now in work) and one trematode specimen now used for molecular analysis. Acquaintance of fellow parasitologists with trematodes from personal collections is also possible.

  1. Taking a count that the present manuscript contains new records, did the material collection follow any ethical procedures? This information could be included by the authors.

Done. The information is presented in Institutional Review Board Statement.

  1. I missed the literature regarding parasite identification in the updated records. All parasites were identified at the species level. Despite being a database, I think that for the updated records, there could be a paragraph talking about species identification; the key used, and which characters were observed which agrees with described species.

We included literature regarding parasite identification in the Methods section.
